# Matryoshka Multimodal Models

**Mu Cai**[1]   **Jianwei Yang**[2]   **Jianfeng Gao**[2]   **Yong Jae Lee**[1]

[1]University of Wisconsin-Madison          [2]Microsoft Research, Redmond

https://matryoshka-mm.github.io/

## Abstract

Large Multimodal Models (LMMs) such as LLaVA have shown strong performance in visual-linguistic reasoning. These models first embed images into a fixed large number of visual tokens and then feed them into a Large Language Model (LLM). However, this design causes an excessive number of tokens for dense visual scenarios such as high-resolution images and videos, leading to great inefficiency. While token pruning and merging methods exist, they produce a single-length output for each image and cannot afford flexibility in trading off information density *v.s.* efficiency. Inspired by the concept of Matryoshka Dolls, we propose $M^3$: *Matryoshka Multimodal Models*, which learns to represent visual content as nested sets of visual tokens that capture information across multiple coarse-to-fine granularities. Our approach offers several unique benefits for LMMs: (1) One can explicitly control the visual granularity per test instance during inference, *e.g.*, adjusting the number of tokens used to represent an image based on the anticipated complexity or simplicity of the content; (2) $M^3$ provides a framework for analyzing the granularity needed for existing datasets, where we find that COCO-style benchmarks only need around 9 visual tokens to obtain an accuracy similar to that of using all 576 tokens; (3) Our approach provides a foundation to explore the best trade-off between performance and visual token length at the sample level, where our investigation reveals that a large gap exists between the oracle upper bound and current fixed-scale representations.

## 1 Introduction

Large Multimodal models (LMMs) (OpenAI, 2023a; Liu et al., 2023a; Zhu et al., 2024; Liu et al., 2024b;a; Wang et al., 2023; Bai et al., 2023) have shown strong performance in visual-linguistic understanding and reasoning. Models such as LLaVA (Liu et al., 2023a; 2024a;b) first embed the input image with a fixed number of visual tokens, and then feed them as prefix tokens to a Large Language Model (LLM) (Vicuna, 2023; Meta, 2024) to reason about the input image. Similar model designs are borrowed in video LMMs (Lin et al., 2023b; Zhang et al., 2023a), where each frame contributes a fixed number of tokens to form the final video representation.

In reality, the number of visual tokens can be prohibitively large in the case of high-resolution images, and even more so for long videos. Existing works (Lin et al., 2023b; Liu et al., 2024b; Zhang et al., 2024b; Team, 2024) mainly tackle this issue by increasing the input context length and consequently, feeding a large number *e.g.,* 3-8k of visual tokens into the LLM. This approach has a couple of significant drawbacks: (1) the extremely long context makes both training and inference inefficient; (2) an excessive number of visual tokens can actually *harm* the LMM's performance, distracting it from attending to the relevant information, as we show in Sec. 4.3. Several recent works (Bolya et al., 2023; Chen et al., 2024; Shang et al., 2024) use heuristics to prune and merge visual tokens to reduce the sequence length. However, they produce a single-length output and *do not afford control over the final sequence length*, which could be useful to trade information density versus efficiency while accounting for resource constraints in the deployment phase.

Images and videos naturally exhibit a hierarchical structure from coarse to fine details, and our human visual system has evolved to recognize visual information in this coarse to fine manner, as shown by biologists and psychologists decades ago (Harris & Giachritsis, 2000; Hegdé, 2008). Can we create a similar structure for LMMs, where within one suite of model weights, the visual content tokens are organized into different scales of granularities? Conceptually, our goal is to learn the visual

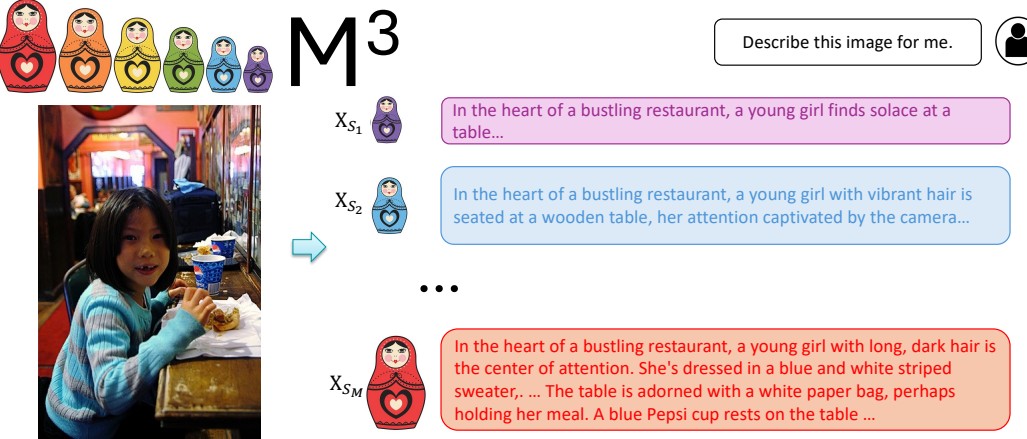

Figure 1: **Matryoshka Multimodal Models.** We enforce the coarser set of visual tokens $\mathbf{X}_{S_{i-1}}$ to be derived from the finer level of visual tokens $\mathbf{X}_{S_i}$. As a result, the granularity of Matryoshka visual tokens gradually changes in a controllable manner. The image is from MSCOCO (Lin et al., 2014) validation set and the captions are generated given 1, 9, and 576 tokens, respectively.

tokens to have a nested structure, similar to the Matryoshka Doll (Kusupati et al., 2022). Matryoshka Representation Learning (MRL) (Kusupati et al., 2022) builds the Matryoshka mechanism over a neural network's representation vector, where each of the segments with various feature dimensions is capable of handling tasks like classification or retrieval. However, for LMMs, the inefficiency mainly comes from the number of tokens. Thus, inspired by, but different from MRL, our work is motivated to build Matryoshka Multimodal Models upon the *token length dimension*, so that we can flexibly adjust it.

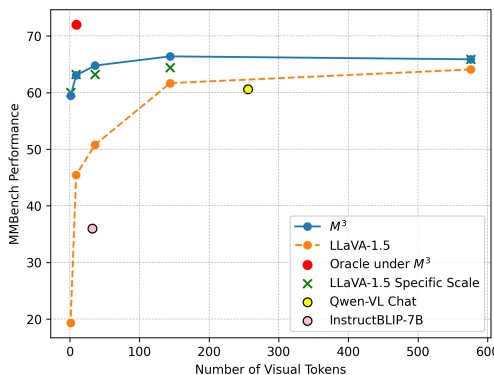

Figure 2: MMBench evaluation results under $M^3$, oracle under LLaVA-1.5-$M^3$, LLaVA-1.5 with average pooling at inference time, LLaVA-1.5 separately trained for each specific scale, and other methods. $M^3$ shows as least as good performance as LLaVA trained for each specific scale. A large gap exists between the oracle upperbound and model's actual performance on a specific scale.

Specifically, we propose $M^3$: *Matryoshka Multimodal Models*, which enforces an LMM to learn a hierarchy of visual representation granularities at the token sequence level, instead of the feature dimension level as in MRL (Kusupati et al., 2022). With this representation, at inference time, the visual granularity can be *flexibly controlled* based on specific requirements, e.g., to account for the input image's information density and efficiency constraints. Our training process is simple and straightforward. During training, we encode the image into $M$ sets of visual tokens from coarse to fine, $\mathbf{X}_{S_i}$, $i = 1, \cdots, M$, where the number of visual tokens gradually increases, *i.e.,* $|\mathbf{X}_{S_{i-1}}| < |\mathbf{X}_{S_i}|$. And importantly, the visual tokens in a coarser level are derived from the visual tokens in a finer level, *i.e.,* $\mathbf{X}_{S_{i-1}} \subset \mathbf{X}_{S_i}, \forall i$. In this way, the visual information in $[\mathbf{X}_{S_1}, \mathbf{X}_{S_2}, \cdots, \mathbf{X}_{S_M}]$ gradually includes more fine-grained details. For example, given a natural image as shown in Figure 1, $\mathbf{X}_{S_1}$ includes high-level semantics such as the restaurant and girl, while $\mathbf{X}_{S_M}$ includes more details such as the Pepsi cup and white paper

bag. All other training settings, such as the loss function and model architecture, are kept the same as LLaVA (Liu et al., 2023a; 2024a;b).

Our approach, $M^3$, introduces several novel properties and benefits for LMMs. First, our approach can efficiently represent visual content where users can flexibly choose the visual token scale at inference time. Under *one suite of weights*, it generates multiple nested sets of visual tokens with different granualarities in information density. This enables flexibility in the number of visual tokens used for any image during inference, enabling control over the best tradeoff between cost and performance

based on the image or video content. For example, one can use all visual tokens for images with dense details and use just a few tokens for simpler images. This flexibility can be particularly significant when handling very long visual sequences, such as videos. For instance, given a fixed budget of 2880 visual tokens, a user could represent a video of 2880 frames each with one token or represent the same video by sampling 5 frames each with 576 tokens.

Second, our method can be used as a general framework to evaluate the visual complexity of vision-language benchmarks, *i.e.,* which level of granularity is needed in order to perform the given task correctly. Surprisingly, we find that most benchmarks, especially those mainly crafted from natural scenes (such as COCO) (Goyal et al., 2017; Li et al., 2023c; Liu et al., 2023b), can be handled well with only $\sim 9$ tokens per image. In contrast, dense visual perception tasks such as document understanding or OCR (Singh et al., 2019; Masry et al., 2022) require a greater amount of tokens $(144 - 576$ tokens) per image to handle the task well. The detailed findings are presented in Sec. 4.2.

Finally, our approach provides a foundation to tackle a critical task in LMMs: *How to use the least amount of visual tokens while answering the visual questions correctly?*. Based on the model's predictions on the test set, we find that compared to full visual tokens, the oracle can use far fewer tokens while performing much better. For example, under six common LMM benchmarks used in LLaVA-NeXT (Liu et al., 2024b), the oracle with the trained $M^3$ model can use as few as 8.9 visual tokens on average to achieve performance that is 8% points better than LLaVA-NeXT which uses 576 tokens per image grid. This indicates that there is a large room for improvement compared to the oracle upperbound, as we show in Sec. 4.2.

To enable further research on adaptive LMMs that learn diverse information granularities, we publicly release our code and models.

## 2 RELATED WORK

**Large Multimodal Models.** Large Language Models (LLMs) like ChatGPT (OpenAI, 2023b), GPT-4 (OpenAI, 2023c), and LLaMA (Touvron et al., 2023) have demonstrated impressive reasoning and generalization capabilities for text. The LLM landscape has been significantly transformed by the introduction of models that also incorporate visual information e.g., GPT-4V (OpenAI, 2023a). Building upon open-source LLMs (Touvron et al., 2023; Vicuna, 2023), a plethora of multimodal models have made significant strides, spearheaded by models like LLaVA (Liu et al., 2023a; 2024a) and MiniGPT-4 (Zhu et al., 2024), which combine LLaMA's (Touvron et al., 2023) language capabilities with a CLIP (Radford et al., 2021) image encoder. Recently, LMMs on more tasks and modalities have emerged, such as region level LMMs (Cai et al., 2024; Zhang et al., 2023c; Chen et al., 2023; Peng et al., 2023; Zhang et al., 2023b), 3D LMMs (Hong et al., 2023), and video LMMs (Lin et al., 2023b; Zhang et al., 2023a; 2024b). However, existing LMMs typically represent the visual content with a large, fixed number of tokens, making it challenging to scale to very long visual sequences such as high-resolution images or long-form videos. In this work, we propose to efficiently represent the visual content by learning multiple nested sets of visual tokens, providing flexibility in the number of visual tokens used for any image during inference.

**Matryoshka Representation Learning.** Matryoshka Representation Learning (MRL) (Kusupati et al., 2022) addresses the need for flexible representations that can adapt to multiple downstream tasks with varying computational resources. This approach, inspired by the nested nature of Matryoshka dolls, encodes information at different granularities within the same high-dimensional feature vector produced by a neural network. The adaptability of MRL extends across different modalities, including vision (ResNet (He et al., 2016), ViT (Dosovitskiy et al., 2021)), vision + language (ALIGN (Jia et al., 2021)), and language (BERT (Devlin et al., 2018)), demonstrating its versatility and efficiency. Recent work (Li et al., 2024) extends MRL to both the text embedding space and the Transformer layers space. Our approach is inspired by MRL, but instead of learning multiple nested embeddings for a high-dimensional feature vector, we learn *nested visual tokens along the token length dimension* for the visual input. We are the first to show that the idea of Matryosha learning can enable explicit control over the visual granularity of the visual content that an LMM processes.

**Token Reduction.** One of the main causes of inefficiency in recent LMMs is their large number of prefix visual tokens that are fed into the LLM (Liu et al., 2023a; Zhu et al., 2024). The quadratic complexity in Transformers (Vaswani et al., 2017) is the key issue in scaling the input sequence length for Transformers. Token reduction serves as an effective technique to reduce computational costs in

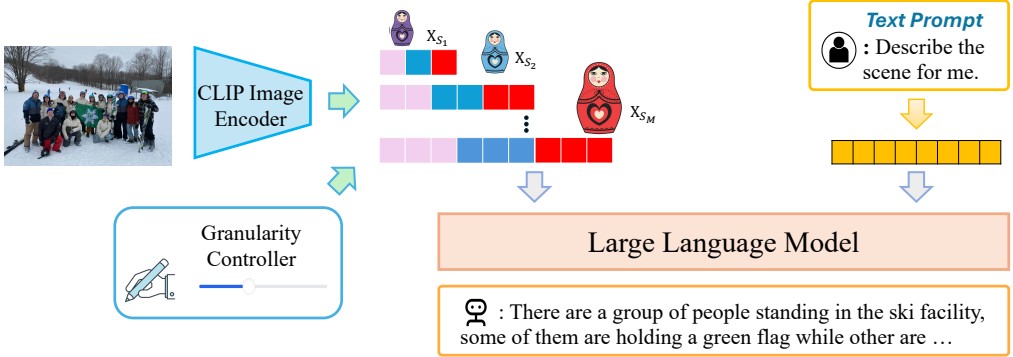

Figure 3: **Architecture of our proposed Matryoshka Multimodal Models.** The visual features from CLIP are represented as several groups of coarse-to-fine visual tokens. At test time, users can explicitly control the granularity of the visual features.

Transformers. Sparse attention methods such as Linformer (Wang et al., 2020) and ReFormer (Kitaev et al., 2020) conduct attention operations within local windows rather than the full context, thereby reducing the quadratic complexity of the vanilla attention operation. Another notable method is Token Merging (ToMe) (Bolya et al., 2023), which utilizes full attention but gradually reduces the number of tokens in each transformer block by selecting the most representative tokens through bipartite matching for the Vision Transformer (ViT). A recent work (Haurum et al., 2023) further studies different families of token reduction methods for ViT. However, prior approaches produce a single length output per input image and do not offer multiple granularities over the reduced token sequence. Our $M^3$ approach instead learns a multi-granularity, coarse-to-fine token representation within the same model architecture and weights, enabling it to easily be adjusted to various computational or memory constraints.

A concurrent work (Hu et al., 2024) shares a similar spirit with our approach, representing an image with varying numbers of visual tokens using a single set of model weights. While their method reformats the visual tokens into a sequential list via transformation layers, we use average pooling to preserve the spatial structure of the visual tokens, demonstrating effectiveness in our experiments.

## 3   $M^3$: Matryoshka Multimodal Models

Our goal is to learn a Large Multimodal Model (LMM) that represents visual content as nested sets of visual tokens capturing information across multiple coarse-to-fine granularities, so that one can explicitly control the visual granularity per test instance during inference. Here we introduce how we learn a Matryoshka doll-like token sequence.

LMMs such as LLaVA (Liu et al., 2023a) typically input a sequence of visual tokens as prefix tokens to the LLM for visual-linguistic reasoning. The visual encoder from pretrained vision-language models, such as CLIP (Radford et al., 2021) and SigLIP (Zhai et al., 2023), is typically utilized to project the images into the set of visual tokens. In particular, the CLIP visual encoder represents an input image $\mathbf{I}$ as an $H \times W$ grid of visual tokens $\mathbf{X}_{H \times W}$, where each $\mathbf{X}_i \in \mathbb{R}^C$ is a $C$ dimensional feature vector. Our goal is to learn nested sets of visual tokens $[\mathbf{X}_{S_1}, \mathbf{X}_{S_2}, \cdots, \mathbf{X}_{S_M}]$ which encode the visual information in a coarse-to-fine manner. To this end, we enforce $\mathbf{X}_{S_i} \subset \mathbf{X}_{S_{i+1}}, \forall i$. Importantly, we do not introduce any new learnable parameters to the LMM. We instead optimize the CLIP visual encoder to learn the nested visual representation directly, and train the ensuing LLM to adapt to the learned nested set of tokens.

For ease of exposition, we consider CLIP-ViT-L-336 (Radford et al., 2021) as the visual encoder, where an image is encoded as $24 \times 24$ visual tokens (576 total). We create $M$ sets of tokens e.g., $|S_i| \in \{1, 9, 36, 144, 576\}$, in which the visual tokens at the coarser level are derived directly from those at the finer level. Specifically, given the initial $24 \times 24$ visual tokens, We sequentially apply $2 \times 2$ pooling with a stride 2, resulting in $12 \times 12$, $6 \times 6$, and $3 \times 3$ visual tokens. Finally, we apply $3 \times 3$ pooling and get the most condensed single visual token. In this way, the sets of Matryoshka visual tokens can gradually preserve the spatial information in the original tokens while simultaneously forming a coarse-to-fine nested representation.

We train $M^3$ by averaging the autoregressive next token prediction loss for each scale $S_i$ for each image $\mathbf{I}_i$. Specifically, given a Matryoshka visual representation $\mathbf{X}_{S_i}$ for scale $S_i$, we maximize the likelihood of the predicted tokens matching the ground-truth answer $\mathbf{X}_a$:

$$P(\mathbf{X}_a \mid \mathbf{X}_{S_i}, \mathbf{X}_q) = \prod_{j=1}^{L} P_{\boldsymbol{\theta}}(x_j \mid \mathbf{X}_{S_i}, \mathbf{X}_q, \mathbf{X}_{a,<j}), \tag{3.1}$$

where $\boldsymbol{\theta}$ is the trainable parameters of the model, which includes both the CLIP visual encoder and the ensuing LLM. $\mathbf{X}_q$ denotes the question in text format, $L$ denotes the token length of the ground truth answer $\mathbf{X}_a$, and $\mathbf{X}_{a,<j}$ denotes all the ground truth answer tokens before the current prediction token $x_j$, where $j$ denotes the token index during text token generation. We omit system messages for clarity, though they are part of the conditioning. Figure 3 shows our model architecture.

The final objective averages over all $M$ visual token scales:

$$\min_{\boldsymbol{\theta}} \frac{1}{M} \sum_{i=1}^{M} -\log P(\mathbf{X}_a \mid \mathbf{X}_{S_i}, \mathbf{X}_q). \tag{3.2}$$

With this objective function, $M^3$ learns nested sets of visual tokens that gradually include more details with increasing scale. For example, in Figure 1, the smaller set of visual tokens describes the whole scene at a high level while the larger set of visual tokens includes more details such as the Pepsi cup. Our training objective affords our model to conduct visual question answering under any granularity during inference. This can be particularly useful in resource constrained applications; e.g., the visual granularity can be flexibly adjusted based on the anticipated simplicity or complexity of the visual content while taking into account compute and memory constraints.

**Discussion.** Given the diverse question-answering training data across multiple domains and sources, the intermediate visual representation in $M^3$ will be forced to represent the visual input to make the prediction be as accurate as possible. Given a small budget such as one token, the model will automatically represent the most salient image attributes in an unsupervised manner.

At a high level, the aforementioned learning mechanism is analogous to that in autoencoder (Hinton & Salakhutdinov, 2006; Coates et al., 2011), which reconstructs the input from the compressed latent code in an unsupervised manner. If the autoencoder has a small/large dimensional latent space, then it will produce a blurrier/sharper reconstruction. But in either case, the learned bottleneck will still try to compress the important information in the input, just to varying degrees.

This is also demonstrated in the prior work Matryoshka Representation Learning (MRL) (Kusupati et al., 2022), where the model can learn the correct granularity even when the supervision signals come from the same single ground-truth target. As mentioned in MRL, the coarse-to-fine granularity is achieved by explicitly optimizing $O(\log(d))$ lower-dimensional vectors in a nested manner, which is analogous to our nested token learning. Empirically, this is further validated by our hallucination analysis on POPE and RefCOCO in Sec. 4.4.

Finally, we argue that it would be fundamentally ill-posed to curate different ground-truth targets for different granularities, as we do not know the criterion of how the different targets should differ. Therefore, we choose to let the multimodal model learn the granularity in an "unsupervised" manner.

## 4 EXPERIMENTS

In this section, we first detail the experiment settings in Sec 4.1. Then we show the performance of $M^3$ on both image-level benchmarks 4.2 and video-level benchmarks 4.3. Finally, we analyze the behavior of Matryoshka Multimodal Models and provide ablations in Sec 4.4 and 4.5.

### 4.1 EXPERIMENT SETTINGS

**Model** We use LLaVA-1.5 (Liu et al., 2024a) and LLaVA-NeXT (Liu et al., 2024b) as the base LMMs, both with Vicuna 7B as the language model backbone. We finetune the whole model using the exact visual instruction data from LLaVA-1.5 and LLaVA-NeXT, respectively. The learning rate of LLM is $2 \times 10^{-5}$ and $1 \times 10^{-5}$, respectively for LLaVA-1.5 and LLaVA-NeXT. The learning

Table 1: Comparison between LLaVA-1.5-$M^3$ across various benchmarks under image understanding benchmarks. LLaVA-1.5-$M^3$ maintains the performance of LLaVA-1.5 while outperforming Qwen-VL and InstructBLIP with fewer tokens.

| Approach | # Tokens | MMBench | GQA | POPE | VizWiz | SEEDBench |
|---|---|---|---|---|---|---|
| Qwen-VL (Bai et al., 2023) | 256 | 38.2 | 59.3 | - | 35.2 | 56.3 |
| Qwen-VL-Chat (Bai et al., 2023) | 256 | 60.6 | 57.5 | - | 38.9 | 58.2 |
| InstructBLIP-7B (Dai et al., 2023) | 32 | 36.0 | 49.2 | - | 34.5 | 53.4 |
| InstructBLIP-13B (Dai et al., 2023) | 32 | - | 49.5 | 78.9 | 33.4 | - |
| LLaVA-1.5-7B | 576 | 64.8 | **62.0** | 85.9 | 54.4 | 60.5 |
| LLaVA-1.5-$M^3$ | 576 | 65.9 | 61.9 | **87.4** | **54.9** | **60.6** |
| | 144 | **66.4** | 61.3 | 87.0 | 53.1 | 59.7 |
| | 36 | 64.8 | 60.3 | 85.5 | 52.8 | 58.0 |
| | 9 | 63.1 | 58.0 | 83.4 | 51.9 | 55.4 |
| | 1 | 59.5 | 52.6 | 78.4 | 49.4 | 50.1 |

rate for the visual encoder is $2 \times 10^{-5}$ for both models. We train both models for 1 epoch using 8 NVIDIA H100 GPUs.

Instead of training the language model from scratch, we initialize the language model weights from pre-trained LLaVA-1.5 and LLaVA-NeXT, which we empirically works better. We name our Matryoshka Multimodal Models LLaVA-1.5-$M^3$ and LLaVA-NeXT-$M^3$.

**Visual Token Scales** We design 5 scales for the visual tokens. LLaVA-1.5 (Liu et al., 2024a) and LLaVA-NeXT (Liu et al., 2024b) both leverage CLIP-ViT-L-336 (Radford et al., 2021) as the visual encoder, where an image is embedded into $24 \times 24$ visual tokens. We gradually apply $2 \times 2$ pooling with stride 2, resulting in $12 \times 12, 6 \times 6$, and $3 \times 3$ visual tokens, where we finally apply a $3 \times 3$ pooling to get the final single visual token. Therefore, the size of Matryoshka visual token sets are $S \in \{1, 9, 36, 144, 576\}$, following a nested manner. The efficiency anlaysis on the system level is shown in Appendix B, where $M^3$ boosts the speed of the LMM prefill process through diminished floating-point operations (FLOPs) and lessens computational memory requirements.

**Evaluations.** For **image understanding**, we evaluate LLaVA-1.5 and LLaVA-NeXT on (a) diverse multimodal benchmarks: POPE (Li et al., 2023c), GQA (Hudson & Manning, 2019), MMBench (Liu et al., 2023b), VizWiz (Gurari et al., 2018), SEEDBench (Li et al., 2023a), ScienceQA (Lu et al., 2022), MMMU (Yue et al., 2024), and (b) document understanding/Optical character recognition (OCR) benchmarks: DocVQA (Mathew et al., 2021), ChartQA (Masry et al., 2022), AI2D (Kembhavi et al., 2016) and TextVQA (Singh et al., 2019).

For **video understanding**, we use both (a) open ended video question answering benchmarks evaluated by GPT-3.5: MSVD-QA (Xu et al., 2017), MSRVTT-QA (Xu et al., 2017) and ActivityNet-QA (Yu et al., 2019); and (b) multi-choice video question answering benchmarks: NExT-QA (Xiao et al., 2021), IntentQA (Li et al., 2023b), and EgoSchema (Mangalam et al., 2024).

### 4.2 IMAGE UNDERSTANDING

**LLaVA-1.5-$M^3$** We evaluate LLaVA-1.5-$M^3$ on the common multimodal understanding and reasoning benchmarks. Results are shown in Table 1. LLaVA-1.5-$M^3$ with full tokens maintains the performance of LLaVA-1.5 across diverse benchmarks. More importantly, our approach shows strong performance even with 1 or 9 tokens. Specifically, in MMBench, a comprehensive multimodal understanding benchmark, LLaVA-1.5-$M^3$ with 9 tokens surpasses Qwen-VL-Chat with 256 tokens, and achieves similar performance as Qwen-VL-Chat with even 1 token. Compared with InstructBLIP (Dai et al., 2023), LLaVA-1.5$M^3$ with 9 tokens surpasses InstructBLIP-7B and InstructBLIP-13B across all benchmarks. This demonstrates that our model has both flexibility and strong empirical performance under diverse number of visual tokens.

**LLaVA-NeXT-$M^3$** We use the proposed Matryoshka Multimodal Models to finetune LLaVA-NeXT, and compare LLaVA-NeXT-$M^3$ with *SS*, which denotes the setting where the LLaVA-NeXT is trained under a **S**pecific **S**cale of visual tokens also for 1 epoch. We also include the oracle upperbound performance: 'Oracle' denotes the case where the best tradeoff between visual tokens

Table 2: Comparison of approaches with the *SS* baseline and $M^3$ across various benchmarks under LLaVA-NeXT (Liu et al., 2024b). Here # Tokens denotes the number of visual tokens per image grid in LLaVA-NeXT. *SS* denotes the baseline model trained with a **S**pecific **S**cale of visual tokens. $M^3$ is at least as good as *SS*, while performing better on tasks such as TextVQA, ChartQA, and MMBench. Oracle denotes the case where the best tradeoff between visual tokens and performance is picked.

| # Tokens Per Grid | Approach | TextVQA | AI2D | ChartQA | DocVQA | MMBench | POPE | ScienceQA | MMMU |
|---|---|---|---|---|---|---|---|---|---|
| 576 | *SS* | 64.53 | 64.83 | 59.28 | 75.40 | 66.58 | 87.02 | 72.29 | 34.3 |
|  | $M^3$ | 63.13 | 66.71 | 58.96 | 72.61 | 67.96 | 87.20 | 72.46 | 34.0 |
| 144 | *SS* | 62.16 | 65.77 | 55.28 | 67.69 | 67.78 | 87.66 | 72.15 | 36.4 |
|  | $M^3$ | 62.61 | 68.07 | 57.04 | 66.48 | 69.50 | 87.67 | 72.32 | 36.1 |
| 36 | *SS* | 58.15 | 65.90 | 45.40 | 56.89 | 67.01 | 86.75 | 71.87 | 36.2 |
|  | $M^3$ | 58.71 | 67.36 | 50.24 | 55.94 | 68.56 | 87.29 | 72.11 | 36.8 |
| 9 | *SS* | 50.95 | 65.06 | 37.76 | 44.21 | 65.29 | 85.62 | 72.37 | 36.8 |
|  | $M^3$ | 51.97 | 66.77 | 42.00 | 43.52 | 67.35 | 86.17 | 71.85 | 35.2 |
| 1 | *SS* | 38.39 | 63.76 | 28.96 | 33.11 | 61.43 | 82.83 | 72.32 | 35.3 |
|  | $M^3$ | 38.92 | 64.57 | 31.04 | 31.63 | 62.97 | 83.38 | 71.19 | 34.8 |
| Oracle | # Tokens | 31.39 | 11.54 | 41.78 | 64.09 | 8.90 | 6.08 | 7.43 | 22.85 |
|  | Performance | 70.51 | 76.36 | 70.76 | 81.73 | 74.35 | 94.29 | 76.07 | 50.44 |

Table 3: Overall accuracy of LLaVA-NeXT-$M^3$ and recent video LMMs on various video understanding benchmarks. Here # Tokens denotes the overall number of visual tokens across all frames.

| Approach | # Tokens | MSVD | MSRVTT | ActivityNet | NextQA | IntentQA | EgoSchema |
|---|---|---|---|---|---|---|---|
| Video-LLaMA (Zhang et al., 2023a) | 32 | 51.6 | 29.6 | 12.4 | - | - | - |
| LLaMA-Adapter (Zhang et al., 2024a) | - | 54.9 | 43.8 | 34.2 | - | - | - |
| Video-ChatGPT (Maaz et al., 2023) | 264+ | 64.9 | 49.3 | 35.2 | - | - | - |
| Video-LLaVA (Lin et al., 2023a) | 2048 | 70.7 | 59.2 | 45.3 | - | - | - |
| InternVideo (Wang et al., 2022) | - | - | - | - | 59.1 | - | 32.1 |
| LLaVA-NeXT-7B (Liu et al., 2024b) | 2880 | 78.8 | 63.7 | 54.3 | **63.1** | **60.3** | 35.8 |
| LLaVA-NeXT-7B-$M^3$ | 2880 | 78.2 | **64.5** | 53.9 | **63.1** | 58.8 | 36.8 |
|  | 720 | **79.0** | **64.5** | 55.0 | 62.6 | 59.6 | 37.2 |
|  | 180 | 77.9 | 63.7 | 55.0 | 61.4 | 59.3 | 37.6 |
|  | 45 | 75.8 | 63.0 | 53.2 | 59.5 | 58.7 | **38.8** |
|  | 5 | 73.5 | 62.7 | 50.8 | 56.5 | 56.7 | 36.2 |

and performance is picked for each test instance; i.e., for each test instance, we select the scale with the fewest amount of tokens but can answer the question correctly. Table 2 shows that our approach, $M^3$, is at least as good as *SS*, while performing better on tasks such as document understanding (TextVQA and ChartQA) and common benchmarks such as MMBench (Liu et al., 2023b).

Our results also show that dataset level biases towards the visual token scales do exist. For example, ScienceQA maintains consistent performance across all visual token scales. AI2D and MMBench only encounter a small performance drop for even as few as 9 to 1 tokens. On the other hand, dense visual perception tasks such as TextVQA and DocVQA show a significant performance drop with fewer tokens. This analysis shows that $M^3$ could serve as a framework to analyze the granularity that a benchmark needs.

Furthermore, there is a large gap between the model's actual performance under full tokens and the upper-bound oracle. This indicates that using full tokens cannot always result in the optimal performance for all samples; i.e., there is a large room of improvement towards the oracle point.

## 4.3 VIDEO UNDERSTANDING

Following IG-VLM (Kim et al., 2024), we directly conduct zero-shot inference on diverse video benchmarks using LLaVA-NeXT-$M^3$. Specifically, 6 frames are uniformly sampled over the entire video, then arranged as a collage, which is fed into LLaVA-NeXT along with the question to get the response. Results under LLaVA-NeXT-$M^3$ and recent video LMMs are show in Table 3.

LLaVA-NeXT-$M^3$ with full visual tokens again shows comparable performance with LLaVA-NeXT. More interestingly, results indicate that full visual tokens usually *do not lead to the best performance* in video understanding tasks. Specifically, on 4 out of 6 benchmarks, full visual tokens show less

Table 4: Comparison between $M^3$ and heuristics based sampling baselines (average pooling, spatial sampling, sequential sampling) at inference time on MMBench with the LLaVA-NeXT architecture.

| # Tokens | $M^3$ | Average Pooling | Spatial Sampling | Sequential Sampling |
|---|---|---|---|---|
| 576 | 67.96 | 67.18 | 67.18 | 67.18 |
| 144 | 69.50 | 61.68 | 65.81 | 60.14 |
| 36 | 68.56 | 50.77 | 60.05 | 44.76 |
| 9 | 67.35 | 45.45 | 45.45 | 31.96 |
| 1 | 62.97 | 19.33 | 26.29 | 22.42 |

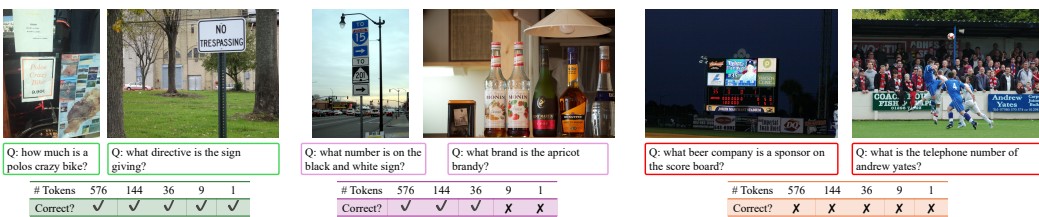

Figure 5: TextVQA test samples with correct and incorrect predictions upon different scales. Answers vary with different number of visual tokens. In addition, $M^3$ can serve as a framework to evaluate the complexity of images.

desirable performance compared to 720 or 180 visual tokens. We suspect that very long visual context could bring distraction (e.g., too much focus on potentially irrelevant background) to the model's prediction, where a compact representation of the video focusing on the more relevant information may be more advantageous.

Finally, for most video understanding tasks such as ActivityNet, IntentQA and EgoSchema, with 9 tokens per image grid (45 tokens in total), the accuracy difference compared to full tokens (2880 in total) is less than 1%. This demonstrates that the video questions in these benchmarks usually require very sparse visual information, as the source of such video understanding benchmarks mostly comes from natural scenes, which matches our observation in image understanding benchmarks.

## 4.4 IN-DEPTH ANALYSIS

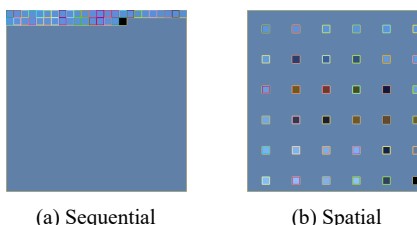

(a) Sequential      (b) Spatial

Figure 4: **Visualization of sequential vs spatial sampling.** Given $24 \times 24$ grids, visualized cells denote sampled tokens.

$M^3$ **shows much stronger performance compared to heuristics based sampling at test time.** A simple way to reduce the number of visual tokens via a training-free way is to conduct heuristic token merging or reduction, which is utilized in works such as Zhang et al. (2024b) and Rippel et al. (2014). In Table 4, we compare $M^3$ with three training-free approaches: average pooling, spatial sampling, and sequential sampling. $M^3$ is much more resilient when the number of tokens decreases, while the heuristic based sampling approaches show dramatic performance drop. A visualization of the spatial and sequential sampling is shown in Figure 4.

$M^3$ **serves as a good metric for image complexity.** We extract the response from LLaVA-NeXT-$M^3$ in the TextVQA benchmark, and show samples where using visual tokens across different scales can answer the question correctly and incorrectly. Shown in Fig. 5, the OCR performance aligns with the complexity of images, indicating that $M^3$ can be utilized as a metric towards sample level complexity.

**Large gap between oracle and actual performance.** As shown in Table 2, the oracle upper-bound can use very few ($6 \sim 64$) tokens yet achieve at least 10% better performance compared to full visual tokens. This suggests that a visual token scale predictor, where the model learns to automatically select the best visual token scale given the input images or both input images and questions, has potential to achieve a better tradeoff. This would be interesting future work.

**Zero-shot generalization to longer visual sequences.** Here we extend the length of the visual tokens at inference time to study the model's zero-shot generalization behavior. Results under LLaVA-NeXT are shown in Table 5. Here LLaVA-NeXT-$M^3$ is trained on $2 \times 2$ image grids but evaluated

Table 5: Performance comparison of different image grid configurations with LLaVA-NeXT-$M^3$.

| # Grids | # Tokens per grid | Overall # Tokens | TextVQA | AI2D | ChartQA | DocVQA | MMBench | POPE | ScienceQA |
|---------|-------------------|------------------|---------|------|---------|--------|---------|------|-----------|
| $2 \times 2$ | 144 | 720 | 62.61 | 68.07 | 57.04 | 66.48 | 69.50 | 87.67 | 72.32 |
| $3 \times 3$ | 144 | 1440 | 64.73 | 67.75 | 58.84 | 70.59 | 69.50 | 87.67 | 72.22 |
| $2 \times 2$ | 576 | 2880 | 63.13 | 66.71 | 58.96 | 72.61 | 67.96 | 87.20 | 72.46 |

Table 6: Accuracy, Precision, and Recall on the POPE dataset with $M^3$.

| # Tokens | Accuracy | Precision | Recall |
|----------|----------|-----------|--------|
| 576 | 88.08 | 94.13 | 81.22 |
| 144 | 88.43 | 93.84 | 82.27 |
| 36 | 88.02 | 88.02 | 82.24 |
| 9 | 87.03 | 92.31 | 80.80 |
| 1 | 83.58 | 83.58 | 82.38 |

Table 7: MMHal-Bench: Score for Adversarial Type and Overall Hallucination Rate.

| # Tokens | Score (Adversarial) | Hallucination Rate |
|----------|---------------------|--------------------|
| 576 | 2.00 | 69% |
| 144 | 2.08 | 56% |
| 36 | 1.92 | 77% |
| 9 | 2.00 | 64% |
| 1 | 2.08 | 69% |

on $3 \times 3$ grids. We set the number of visual tokens to be 144 in each image during evaluation. The model obtains a significant improvement in document understanding by 2.12, 1.80, and 4.11 on TextVQA, ChartQA, and DocVQA, respectively, while maintaining the same performance on benchmarks mainly composed of natural scene images. $3 \times 3$ image grids with 144 tokens per grid own 1440 tokens, yet achieve similar performance with the default LLaVA-NeXT $2 \times 2$ image grids with 2880 total tokens (576 tokens per grid). This indicates it is promising to feed more subimages while making the number of visual tokens within each subimage much smaller.

**Detailed investigation into potential hallucinations.** We next investigate potential hallucinations of $M^3$ on two popular benchmarks: POPE (Li et al., 2023c) and MMHal-Bench (Sun et al., 2023). Table 6 shows results on POPE. Accuracy reflects the proportion of correctly answered questions, Precision and Recall reflect the ratios of correctly answering questions whose answers are "Yes" or "No", respectively. $M^3$ maintains the same level of recall while showing gradually increasing precision as the number of tokens is increased. Results for MMHal-Bench (Table 7) reveal similar trends. Different numbers of visual tokens show similar performance on the "adversarial" type. Similar to the "No" question type in POPE, here "adversarial" questions ask models about the objects that do not exist in the image. Thus "adversarial" questions expects models to say "No". Different numbers of visual tokens show similar overall hallucination rate. Overall, these results demonstrate that $M^3$ maintains a similar level of hallucination across different scales of visual tokens.

**Effect of object size.** For the objects appearing in POPE images, we categorize the objects into three groups based on the area occupied by the object. With more visual tokens, $M^3$ shows significant performance boost in small objects compared to the large objects; see Table 8. Visual grounding results on RefCOCO testA set (ACC@0.5) (Yu et al., 2016) corroborate this finding, where more visual tokens leads to significant performance boost for understanding small objects compared to large objects; see Table 9. These results indicate that $M^3$ tends to preserve more coarse-grained information when the number of tokens are limited. Importantly, both the hallucination experiments from the previous section and the results here demonstrate that coarse-to-fine granularities can be learned in an unsupervised manner, using one ground-truth target for all token lengths.

## 4.5 ABLATION STUDIES

**Matryoshka visual token sampling.** Here we compare three different ways to select the visual tokens for Matryoshka Multimodal Models, including average pooling, spatial sampling, and sequential sampling, which is illustrated in Figure 4. Shown in Table 10, averaging pooling shows better performance than the two alternatives across diverse benchmarks. In general, sequential sampling performs the worst. We hypothesize that this is due to the visual tokens having spatial information, while sequential sampling does not naturally align with the spatial distribution of the visual tokens.

**Training the entire LMM vs only training CLIP.** Since the nested behavior of Matryoshka visual tokens is learned within the CLIP visual encoder, we next evaluate whether it is necessary to also finetune the LLM. Shown in Table 11, training the whole LLM achieves better performance. This demonstrates that by also training the LLM, the model can better adapt to the patterns of the visual tokens distributed in the Matryoshka manner.

Table 8: $M^3$ Performance on different scales with varying numbers of visual tokens on POPE.

| # Tokens | Small | Medium | Large |
|---|---|---|---|
| 576 | 85.20 | 84.05 | 88.98 |
| 144 | 86.31 | 86.11 | 88.98 |
| 36 | 78.19 | 82.50 | 87.96 |
| 9 | 76.39 | 80.53 | 85.40 |
| 1 | 78.53 | 80.02 | 85.31 |

Table 9: RefCOCO visual grounding accuracy for different object sizes.

| # Tokens | Small | Medium | Large | Average |
|---|---|---|---|---|
| 576 | 76.2 | 86.4 | 88.5 | 83.7 |
| 144 | 70.3 | 84.5 | 87.3 | 80.7 |
| 36 | 54.8 | 75.4 | 85.5 | 71.9 |
| 9 | 30.5 | 55.9 | 75.2 | 53.9 |
| 1 | 7.8 | 14.2 | 28.8 | 16.9 |

Table 10: Ablation on Matryoshka visual token sampling including average pooling, sequential sampling, and spatial sampling with LLaVA-NeXT-$M^3$.

| Num of Vis Tokens | TextVQA | | | MMBench | | | AI2D | | |
|---|---|---|---|---|---|---|---|---|---|
| | Avg Pooling | Sequential | Spatial | Avg Pooling | Sequential | Spatial | Avg Pooling | Sequential | Spatial |
| 576 | 63.13 | 59.37 | 60.45 | 67.96 | 64.60 | 64.43 | 66.71 | 65.61 | 64.96 |
| 144 | 62.61 | 55.80 | 58.33 | 69.50 | 64.18 | 64.52 | 68.07 | 64.90 | 64.96 |
| 36 | 58.71 | 52.79 | 52.39 | 68.56 | 63.92 | 64.69 | 67.36 | 64.51 | 64.02 |
| 9 | 51.97 | 44.05 | 44.19 | 67.35 | 63.14 | 62.11 | 66.77 | 63.70 | 63.92 |
| 1 | 38.92 | 28.03 | 29.91 | 62.97 | 59.36 | 57.47 | 64.57 | 63.21 | 63.08 |

Table 11: Performance comparison of training LLaVA-NeXT-$M^3$ with and without training the LLM across diverse benchmarks. We see a clear drop when freezing the LLM.

| Num of Vis Tokens | TextVQA | | MMBench | | AI2D | | DocVQA | |
|---|---|---|---|---|---|---|---|---|
| | w/ LLM | w/o LLM | w/ LLM | w/o LLM | w/ LLM | w/o LLM | w/ LLM | w/o LLM |
| 576 | 63.13 | 61.16 | 67.96 | 63.66 | 66.71 | 63.92 | 72.61 | 69.15 |
| 144 | 62.61 | 57.79 | 69.50 | 65.21 | 68.07 | 63.73 | 66.48 | 59.77 |
| 36 | 58.71 | 49.75 | 68.56 | 63.92 | 67.36 | 62.89 | 55.94 | 44.08 |
| 9 | 51.97 | 36.15 | 67.35 | 61.08 | 66.77 | 62.05 | 43.52 | 28.36 |
| 1 | 38.92 | 19.72 | 62.97 | 51.80 | 64.57 | 60.59 | 31.63 | 17.37 |

Table 12: Comparison of performance between average pooling (parameter-free) and learning-based approaches including Resampler (Alayrac et al., 2022), C-Abstractor (Cha et al., 2024), and D-Abstractor (Cha et al., 2024) on MMBench and TextVQA with LLaVA-1.5-$M^3$.

| # Tokens | MMBench | | | | TextVQA | | | |
|---|---|---|---|---|---|---|---|---|
| | Avg Pooling | Resampler | C-Abstractor | D-Abstractor | Avg Pooling | Resampler | C-Abstractor | D-Abstractor |
| 576 | **65.9** | 65.8 | 65.4 | 58.8 | **57.75** | 53.50 | 55.47 | 48.22 |
| 144 | **66.4** | 66.0 | 66.3 | 56.7 | **56.76** | 51.95 | 54.32 | 46.69 |
| 36 | 64.8 | 63.9 | **64.9** | 55.8 | **54.97** | 48.67 | 51.87 | 46.09 |
| 9 | **63.1** | 59.4 | 62.8 | 53.5 | **53.50** | 49.83 | 46.53 | 45.24 |
| 1 | **59.5** | 53.2 | 58.4 | 49.6 | **49.49** | 44.64 | 46.05 | 44.12 |

**Comparison with training-based visual token reduction.** Results on MMBench and TextVQA in Table 12 show that average pooling used in $M^3$ overall produces the best performance compared to three learning based approaches: (a) resampler (Cha et al., 2024), (b) C-Abstractor (Cha et al., 2024), and (c) D-Abstractor (Cha et al., 2024). This is likely because averaging pooling preserves spatial details better. Furthermore, average pooling does not require any new learned layers or parameters.

## 5 CONCLUSION

We introduced $M^3$: Matryoshka Multimodal Models, which learns to represent visual content as nested sets of visual tokens, capturing information across multiple coarse-to-fine granularities. LMMs equipped with $M^3$ afford explicit control over the visual granularity per test instance during inference. We also showed that $M^3$ can serve as an analysis framework to investigate the visual granularity needed for existing datasets, where we discovered that a large number of multimodal benchmarks only need as few as 9 visual tokens to obtain accuracy similar to that of using all visual tokens, especially for video understanding. Furthermore, we disclosed a large performance-efficiency gap between the oracle upper-bound and the model's performance. Our work can be naturally extended to other domains. For example, the long context in a text-only LLM or vision tokens in dense vision tasks can also be represented as nested sets of tokens in a Matryoshka manner.

## 6 REPRODUCIBILITY STATEMENT

We have publicly released our code, data, and pretrained models, so that the community can fully reproduce, and build-upon, our work.

## ACKNOWLEDGEMENT

This work was supported in part by NSF CAREER IIS2150012, NSF IIS-2404180, and Institute of Information & communications Technology Planning & Evaluation(IITP) grants funded by the Korea government(MSIT) (No. 2022-0-00871, Development of AI Autonomy and Knowledge Enhancement for AI Agent Collaboration) and (No. RS2022-00187238, Development of Large Korean Language Model Technology for Efficient Pre-training), and Microsoft Accelerate Foundation Models Research Program.

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

# APPENDIX

## A  BROADER IMPACT

The broader impact of $M^3$, a framework with nested visual representations, has potential benefits and risks associated with its deployment and release. Our model is trained using the exact same architecture and data of LLaVA-1.5 (Liu et al., 2024a) and LLaVA-NeXT (Liu et al., 2024b). All the concerns are same as LLaVA. Specifically, as one example, LLaVA conducts instruction tuning using GPT-4 and GPT-4V generated data. The bias from GPT-4 and GPT-4V would still exist in LLaVA.

## B  EFFICIENCY ANALYSIS

To illuminate the computational benefits conferred by $M^3$, we employ the roofline-based LLM-Viewer analysis as detailed in (Yuan et al., 2024). Our analysis is set within a hypothetical context designed to emphasize the effects of $M^3$ on processing efficiency in LMMs. We study the LLaVA-1.5 case where a $336 \times 336$ resolution image is processed using a CLIP-ViT image encoder, resulting in 576 visual tokens. Accompanied by a text prompt with an assumed number of 30 tokens, the nested visual tokens in $M^3$ substantially lowers the visual token count. The consequences of this reduction are substantial as outlined in Table 13, detailing the computational costs involved in the LMM prefill process. Notably, $M^3$ not only boosts the speed of the LMM prefill process through diminished floating-point operations (FLOPs) but also lessens computational memory requirements.

It is crucial to highlight that the advantages of $M^3$ are not limited to just efficiency improvements. The token reduction approach of $M^3$ can also enhance other LMM acceleration methods, such as quantization and factorization, as referenced in (Yuan et al., 2023). This complementary relationship accentuates the broad potential of $M^3$ to contribute to a wider array of efficiency-boosting strategies.

Table 13: Computation Cost Analysis. The development device is Tesla V100 GPU, and time estimated by the roofline model represents the theoretical performance that the hardware can achieve.

| # Tokens | FLOPs (T) | Prefill Time (ms) | Total Memory (GB) | Storing Activation (GB) |
|---|---|---|---|---|
| 576 | 8.0 | 58.1 | 21.6 | 3.8 |
| 144 | 2.2 | 19.5 | 15.0 | 0.7 |
| 36 | 0.9 | 18.0 | 13.8 | 0.3 |
| 9 | 0.5 | 17.7 | 13.6 | 0.2 |
| 1 | 0.4 | 17.6 | 13.5 | 0.1 |

## C    MORE ABLATION STUDIES: INITIALIZE WEIGHTS FROM LLaVA AND AVERAGE LOSS UPON ALL TOKEN SCALES

As explained in Sec. 3 and 4.1, we (a) initialize the LLM weights from LLaVA and (b) minimize the loss averaged upon all visual token scales for each sample during training. An alternative choice is to randomly sample a visual token scale. Shown in Table 14, initializing the LLM weights from LLaVA and minimizing the losses over all scales shows consistent performance boost compared to using the vanilla text-only pre-trained LLM weights (Vicuna, 2023) and randomly selecting a visual token scale. Initializing the LLM weights from LLaVA makes the training process of $M^3$ more stable. By learning all scales at once, the model is forced to learn the nested behavior for each sample, which leads to better performance.

## D    MORE VISUALIZATIONS ON NESTED VISUAL REPRESENTATION

Shown in Figure 6, with more visual tokens, LMMs can discover more details, such as furniture and human attributes. Besides, LMMs can generate higher quality descriptions with more visual tokens, as demonstrated by the OCR capability in Figure 6 (b).

## E    THE BLIND LOWER BOUND PERFORMANCE WITH **0 VISUAL TOKENS**

Table 15 shows that 0 visual tokens (i.e., a text-only model) align with the trend shown in our paper under different numbers of visual tokens. Removing all visual tokens results in significant performance drop on most benchmarks especially for document understanding such as ChatVQA and DocVQA. On the other hand, ScienceQA, MMMU, and AI2D show very similar performance when dropping all visual tokens, demonstrating that language prior plays a much more significant role in those benchmarks.

## F    PERFORMANCE OF TOKEN CONCATENATION UNDER $M^3$

An intriguing question is: *Can token concatenation under $M^3$ further improve performance?* As shown in Table 16, (1) concatenating all token scales shows slightly better performance but requires many more visual tokens. (2) concatenating all but the last token scale shows comparable performance to the largest visual token scale (576 tokens).

## G    GENERALIZATION TO MORE TOKEN SCALES

We conduct experiments on a set of new scales. As shown in Table 17, on new scales, $M^3$ can still represent images in a nested manner with strong performance.

Table 14: Impact of (a) initializing the LLM weights from LLaVA, and (b) averaging the loss from all scales vs randomly selecting a scale for each sample during training with LLaVA-NeXT-$M^3$.

| Technique | TextVQA | | | | AI2D | | | |
|---|---|---|---|---|---|---|---|---|
| Init LLM weights from LLaVA | | ✓ | | ✓ | | ✓ | | ✓ |
| Average losses over all scales | | | ✓ | ✓ | | | ✓ | ✓ |
| 576 | 60.36 | 62.25 | 61.01 | 63.13 | 62.40 | 65.06 | 65.84 | 66.71 |
| 144 | 59.61 | 61.02 | 59.80 | 62.61 | 63.67 | 65.61 | 65.77 | 68.07 |
| 36 | 54.86 | 55.91 | 55.32 | 58.71 | 63.67 | 65.32 | 66.68 | 67.36 |
| 9 | 46.84 | 47.04 | 48.80 | 51.97 | 63.02 | 64.83 | 65.38 | 66.77 |
| 1 | 33.78 | 33.68 | 36.05 | 38.92 | 61.53 | 63.21 | 63.37 | 64.57 |

Table 15: Performance of $M^3$ across various benchmarks under LLaVA-NeXT (Liu et al., 2024b). Here # Tokens denotes the number of visual tokens per image grid in LLaVA-NeXT, including **0 visual token**.

| # visual tokens | TextVQA | AI2D | ChartQA | DocVQA | MMBench | POPE | ScienceQA | MMMU |
|---|---|---|---|---|---|---|---|---|
| 576 | 63.13 | 66.71 | 58.96 | 72.61 | 67.96 | 87.20 | 72.46 | 34.00 |
| 144 | 62.61 | 68.07 | 57.04 | 66.48 | 69.50 | 87.67 | 72.32 | 36.10 |
| 36 | 58.71 | 67.36 | 50.24 | 55.94 | 68.56 | 87.29 | 72.11 | 36.80 |
| 9 | 51.97 | 66.77 | 42.00 | 43.52 | 67.35 | 86.17 | 71.85 | 35.20 |
| 1 | 38.92 | 64.57 | 31.04 | 31.63 | 62.97 | 83.38 | 71.19 | 34.80 |
| 0 | 10.03 | 60.43 | 14.24 | 11.95 | 21.48 | 56.65 | 70.79 | 31.20 |

Besides, other Matryoshka sampling strategies can also support a diverse number of visual tokens at inference time, such as sequential sampling. However, in this paper, we choose average pooling due to its superior performance, as demonstrated in the ablations in Table 10.

## H    STUDY ON A MORE STRICT ORACLE PERFORMANCE WITH MULTIPLE RUNS

We run evaluations for 5 times using different seeds with temperature 0.2 to get a more strict oracle. Then, we take a majority vote for the prediction for each sample in each scale. Shown in Table 18, with 5 runs, the upper bound performance is similar to that of a single run.

## I    STUDY ON A MORE STRICT ORACLE PERFORMANCE WITH MULTIPLE RUNS

Our proposed $M^3$ is a plug-and-play methodology that can be generally applied to multimodal models including both LLaVA-style (treating visual tokens as prompts) and Flamingo-style (Alayrac et al., 2022) (using cross-attention to fuse visual features into text features). Specifically, Flamingo-style MLLMs (i) first extract visual features using the Perceiver Resampler Alayrac et al. (2022), and (ii) then apply cross attention for multimodal fusion. $M^3$ can be naturally applied between these two stages to produce the multi-granularity visual representation before the cross attention stage, resulting in the Flamingo-$M^3$. Therefore, the computation, reflected by measures such as FLOPS, can also be reduced if a coarse-grained visual representation scale is used.

Specifically, consider $n_V$ as the number of visual tokens, $n'_V$ as the number of visual tokens after passing the perceiver resampler in Flamingo, $n_L$ as the number of text tokens, $M$ as the visual token reduction scale for our proposed $M^3$, $d_V and d_L$ as the hidden dimension in the vision encoder and language model, respectively. For simplicity, we omit the multimodal connector (MLP in LLaVA and Perceiver Resampler in Flamingo), as well as the number of layers in vision encoder and language decoder since they are in the same scale (24 *v.s.* 32 layers in LLaVA-1.5-7B). Then the computational complexity can be expressed in Table 19.

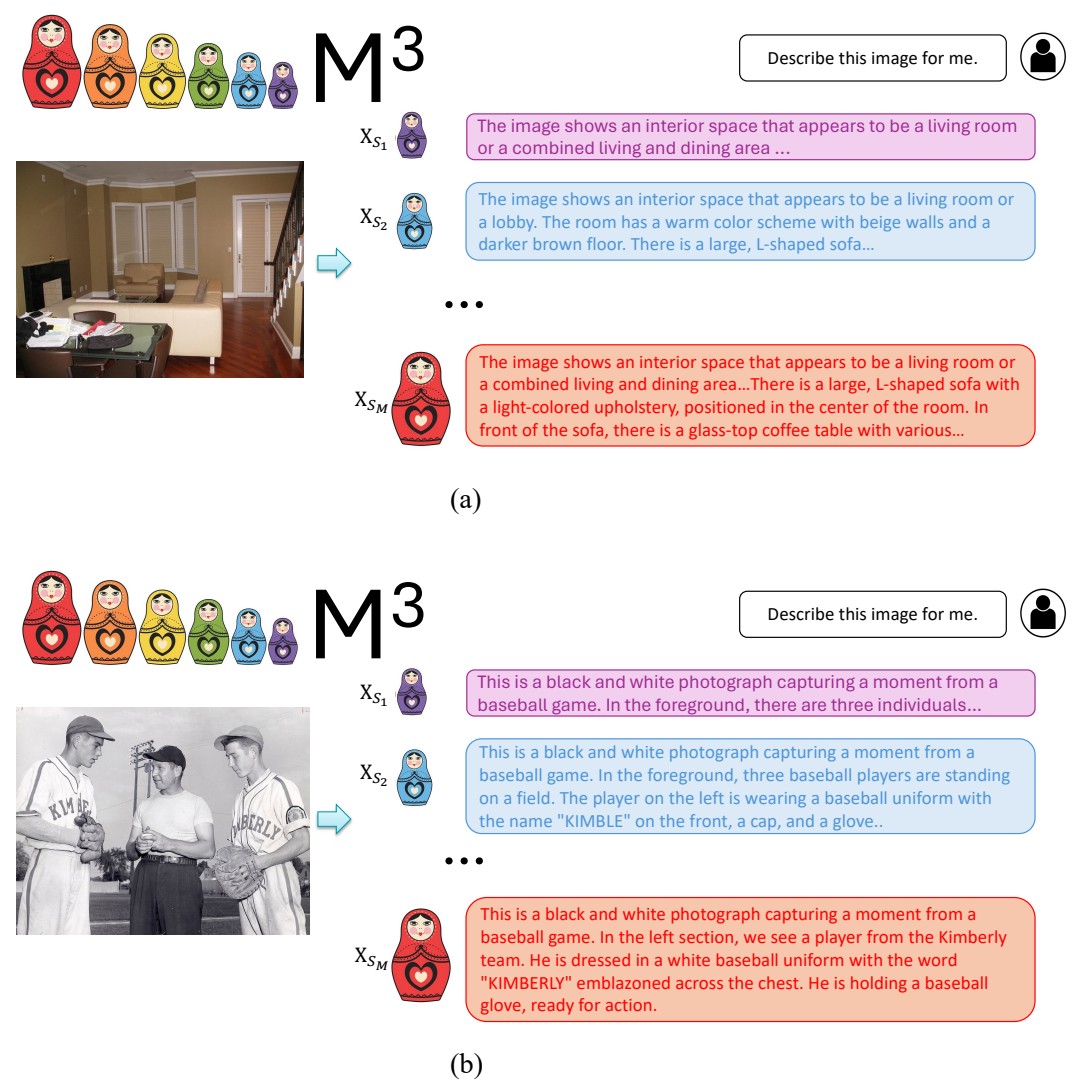

Figure 6: **More visualization examples.** With more visual tokens, LMMs can discover more details, and generate higher quality descriptions. The images are from MSCOCO (Lin et al., 2014) validation set.

Table 16: Performance of $M^3$ across various benchmarks under different numbers of visual tokens and when (1) all token scales are concatenated or (2) all but the last token scale are concatenated.

| # visual tokens | MMBench | GQA | POPE | VizWiz | SEEDBench |
|---|---|---|---|---|---|
| 576 | **65.9** | 61.9 | 87.4 | **54.9** | 60.6 |
| 144 | 66.4 | 61.3 | 87.0 | 53.1 | 59.7 |
| 36 | 64.8 | 60.3 | 85.5 | 52.8 | 58.0 |
| 9 | 63.1 | 58.0 | 83.4 | 51.9 | 55.4 |
| 1 | 59.5 | 52.6 | 78.4 | 50.1 | 49.4 |
| 144+36+9+1=190 | 63.8 | 63.5 | 86.7 | 56.1 | 61.2 |
| 576+144+36+9+1=766 | 62.5 | **63.7** | 85.0 | **56.7** | **61.4** |

Table 17: Performance of $M^3$ across various benchmarks under new sets of visual token scales. On new scales, $M^3$ can still represent the images in a nested manner with strong performance.

| # visual tokens | MMBench | GQA | POPE | VizWiz | SEEDBench |
|---|---|---|---|---|---|
| 576 | 67.4 | 61.5 | 87.2 | 49.5 | 60.2 |
| 144 | 65.2 | 59.3 | 85.3 | 46.5 | 57.1 |
| 64 | 64.7 | 58.4 | 84.3 | 46.0 | 55.1 |
| 36 | 63.2 | 57.0 | 83.6 | 45.9 | 53.1 |
| 16 | 61.8 | 54.7 | 80.9 | 44.7 | 50.2 |
| 9 | 58.7 | 53.3 | 78.4 | 42.7 | 49.1 |
| 4 | 54.0 | 51.3 | 77.5 | 41.4 | 45.2 |
| 1 | 47.9 | 47.7 | 74.6 | 40.3 | 43.1 |

Table 18: Token counts and performance across multiple datasets for 1 run and 5 runs for the oracle performance of $M^3$.

| | TextVQA | AI2D | ChartQA | DocVQA | MMBench | GQA | ScienceQA | MMMU |
|---|---|---|---|---|---|---|---|---|
| **1 run** # Tokens | 31.39 | 11.54 | 41.78 | 64.09 | 8.90 | 6.08 | 7.43 | 22.85 |
| **Oracle Performance** | 70.51 | 76.36 | 70.76 | 81.73 | 74.35 | 94.29 | 76.07 | 50.44 |
| **5 runs** # Tokens | 33.24 | 13.42 | 44.05 | 67.54 | 9.52 | 6.09 | 8.05 | 21.03 |
| **Oracle Performance** | 71.00 | 75.94 | 69.44 | 81.23 | 76.48 | 94.27 | 76.94 | 50.11 |

Table 19: The computational complexity of the original model and $M^3$ in LLaVA and Flamingo style models.

| Model | Original Complexity | Matryoshka Complexity |
|---|---|---|
| LLaVA | $O(n_V^2 \cdot d_V + (n_V + n_L)^2 \cdot d_L)$ | $O(n_V^2 \cdot d_V + (\frac{n_V}{M} + n_L)^2 \cdot d_L)$ |
| Flamingo | $O(n_V^2 \cdot d_V + n_V' \cdot n_L \cdot d_L + n_L^2 \cdot d_L)$ | $O(n_V^2 \cdot d_V + \frac{n_V'}{M} \cdot n_L \cdot d_L + n_L^2 \cdot d_L)$ |

