# OpenReview forum: "Matryoshka Multimodal Models"
_ICLR.cc/2025/Conference — ICLR 2025 Poster_

### Official Review · Reviewer_swzi · 2024-11-02

**Soundness:** 3
**Presentation:** 4
**Contribution:** 3
**Rating:** 6
**Confidence:** 3

**Summary:**

This paper focuses on the VLLM domain and proposes to use average pooling to downsample the visual tokens to a hierarchy of scales for efficient computation.

**Strengths:**

1. Presentation is clear and concise
2. The method is simple but effective
3. Experiments are well-designed and can support the value of the proposed method
4. The large number of visual tokens in multi-modal LLMs is a significant and urgent problem

**Weaknesses:**

1. Whereas the paper claims that the method can ''**adaptively** and efficiently represent visual content", I think the word **adaptively** is kind of misleading here as it makes readers feel that the method per se includes some dynamic inference strategy. I can understand that the authors hope to express that the users can tradeoff computation for accuracy, but it is important to make it more clear.


(**This is actually a weakness of mine instead of the paper**) I am not able to precisely evaluate the novelty of the paper because the VLLM domain is altering from day to day and I have not been closely tracing the frontier papers in the token reduction topic, and I hope other reviews can give better novelty judgments.

**Questions:**

None

---

### Official Review · Reviewer_rbWX · 2024-11-02

**Soundness:** 3
**Presentation:** 3
**Contribution:** 3
**Rating:** 6
**Confidence:** 3

**Summary:**

This paper addresses the problem of visual representation in large multimodal models (LMMs) with a method called Matryoshka Multimodal Models (M3). M3 sequentially applies average pooling to the initial visual tokens extracted by the CLIP-ViT to obtain visual representations at different granularity levels. During training, the LMM learns to autoregressively generate the next tokens based on visual representations at each granularity level individually. In inference, the level of visual granularity can be adjusted to balance performance and efficiency. Experimental results indicate that, on a number of benchmarks, M3 achieves performance on par with LLaVA-1.5 and LLaVA-Next, while requiring significantly fewer visual tokens. Additionally, M3 provides a tool for analyzing the visual complexity of vision-language benchmarks by assessing the granularity required to arrive at correct answers. The results reveal that dense visual perception benchmarks, such as TextVQA and DocVQA, indeed require a higher number of visual tokens compared to other benchmarks.

**Strengths:**

- This paper is well-motivated, addressing the important capability of representing visual information at varying levels of granularity. This flexibility enables adjusting the number of visual tokens based on both computational budget and task complexity.
- The proposed method is effective, demonstrating comparable performance with the baseline LMMs (LLaVA-1.5 and LLaVA-Next) while using significantly fewer visual tokens, across benchmarks that do not demand dense visual perception.
- The empirical study offers several valuable insights:
  - A substantial gap exists between the naive use of all visual tokens and the upper-bound performance achieved by selecting the optimal number of tokens for each test instance.
  - Different vision-language tasks indeed require varying numbers of visual tokens to be addressed.
  - Reducing the number of visual tokens does not increase the level of hallucination for M3.
  - The ablation study compressively compares different token reduction methods and suggests the advantage of average pooling.

**Weaknesses:**

- Although M3 can produce visual representations at multiple granularity levels, the number of visual tokens used at inference must be predefined. In other words, the method cannot adaptively adjust the number of visual tokens for different instances.
- The baseline methods used for video understanding are relatively weak. For example, recent 7B-scale VLMs have achieved over 60% accuracy on EgoSchema, while the best baseline in this work only reaches 35.8%. M3 would likely benefit from integration with more advanced video LMMs. It would also be better to explore alternative video encoding methods other than "arranging video frames into a collage".
- The method for obtaining Oracle performance is not clearly explained. In line 323, the authors state, "for each test instance, we select the scale with the fewest tokens that can answer the question correctly." However, it is unclear what happens if the model cannot produce the correct answer at any scale.
- The capability of the LMMs is not taken into account when using M3 as a tool to analyze the visual complexity of vision-language tasks. If the LMMs cannot effectively consume additional visual information, increasing the number of visual tokens may negatively impact performance, even if the task itself requires more visual information.

**Questions:**

N/A

---

### Official Review · Reviewer_pHpw · 2024-11-04

**Soundness:** 3
**Presentation:** 3
**Contribution:** 2
**Rating:** 6
**Confidence:** 4

**Summary:**

This paper proposed a multimodal LLM M3 supporting varing number of visual tokens, inspired by the Matryoshka Dolls. By changing the nbumber of visual tokens, M3 can understand the visual content at different granularity. The proposed method is simple and easy to follow, which has been verified by the experiments.

**Strengths:**

- The motivation is clear. Current LMMs need more and more visual tokens to enhance their performance, the study of token reduction is important for efficient LMMs.
- The method is simple and easy to implement. Instead of tuning LLM for accepting varing number of tokens, M3 shows that tuning CLIP also works.
- The main evaluation and ablation analysis confirm M3's effectiveness.

**Weaknesses:**

- Comparisions with dynamic sampling methods like Token Merging and Chat-Univi [1]. The performance drop is significant when reducing the number of tokens, while dynamic sampling methods like Chat-Univi can even surpass its full token baseline. Besides, M3 can be regarded as a special case of dynamic sampling. I suggest a fair comparison with these methods.
- High-resolution and long video evaluation and comparisons with other works (LLaVA-HD, SPHINX, LLaMA-VID etc.) . Since these tasks usually requires more tokens, M3 may lead to a better trade-off between performance and computation.
- Suggest to add speed and computation cost comparisions on the main paper, instead of the appendix.

[1] Chat-UniVi: Unified Visual Representation Empowers Large Language Models with Image and Video Understanding

**Questions:**

- How to extend M3 to any number of tokens. Currently, it can only support a set of token numbers (576->144->xxx). If we can enlarge the set, then switching between them will be smoother.

---

### Official Review · Reviewer_8DFt · 2024-11-04

**Soundness:** 4
**Presentation:** 4
**Contribution:** 3
**Rating:** 6
**Confidence:** 4

**Summary:**

The paper proposes Matryoshka Multimodal Models (M^3), a MLLM framework supporting variable visual token length for flexible inference cost control. The multi-scale visual features are generated in a homogeneous manner by pooling visual encoder output with different kernel sizes, and the same set of weights are jointly trained on all scales, producing a single all-in-one model for easy deployment. Models are constructed in a continual finetuning setting on top of LLaVA and LLaVA-NeXT, and evaluated on a broad range of MLLM benchmarks with extensive ablation studies.

**Strengths:**

* The paper tackles a meaningful problem in practice. The motivation is coherent and easy-to-follow.

* The method description is mostly clear.

* Extensive experiments with strong results and insightful analysis.

**Weaknesses:**

* **Comparison with Flamingo-style (i.e., cross-attention-based) methods**: Despite the popularity of LLaVa-style MLLMs which treat visual tokens as prompt, Flamingo-style MLLMs, which decode text-conditioned salient visual features with cross-attention modules, are also studied as an alternative paradigm in several previous works, e.g., [1, 2]. It's noteworthy that cross-attention alleviates most of the the performance penalty due to long visual sequences by nature, because the visual tokens do not go through the expensive MLP and quadratic-complexity self-attention in the language model part (it may still be quadratic in the visual encoder part though). Even if extensive experiments are infeasible within the rebuttal period, it might still make the argument stronger and benefit future readers if at least some discussions could be included (e.g., related works, theoretical analysis of both methods, FLOPS comparison, what if they are used together).

* **Training cost**: From the description at around Line 226, it seems that *all* scales of *all* images are used for training, which means the training cost could be a few times of single-scale training. A comparison with specific numbers and some discussions on scalability would be very helpful as training data of state-of-the-art MLLMs are approaching billion-scale.

* **Minor writing issues**: In Table 13, the unit of FLOPs should be in T instead of TB (TB=TeraBytes is for memory). Repeated reference item: Zhang et al 2023a / Zhang et al 2023b.

[1] Alayrac, Jean-Baptiste, et al. "Flamingo: a visual language model for few-shot learning." Advances in neural information processing systems 35 (2022): 23716-23736.

[2] Zhang, Renrui, et al. "LLaMA-adapter: Efficient fine-tuning of large language models with zero-initialized attention." The Twelfth International Conference on Learning Representations. 2024.

**Questions:**

* **Symbol clarification**: At Line 203, it's stated that $X_{S_i} \subset X_{S_{i + 1}}$. However, if we consider $X_{\cdot}$ as a set of $C$-dimensional features, then this seems self-contradictory because the average of a subset of elements $\frac{1}{n} \sum_j X_{S_{i + 1}, j}$ is not necessarily one of the original elements. Could the authors provide a more rigorous definition about symbols $X_{S_i}$ and, in turn, the definition of Matryoshka property studied in this case?

---

> ### Comment · Reviewer_8DFt · 2024-11-25
>
> Thanks for the responses from the authors as well as the other reviewers. Most of my concerns are well resolved. I would like to maintain the positive score.

---

### Public Comment · ~Mouxing_Yang1 · 2025-02-17
**About the Author Responses**

Hi, congratulations on the acceptance of the paper—it’s an interesting piece of work! However, I can’t find the authors’ response to the reviewers. Is there an issue in the page, or was it not included?

---

### Meta-Review · Area_Chair_gQCB · 2024-12-21

**Metareview:**

This paper introduces Matryoshka Multimodal Models, which make large multimodal models more efficient by representing visual content with adjustable token granularity. The strengths include clear motivation, simple but effective methods, and solid experiments showing flexibility and efficiency. Weaknesses are limited runtime adaptability and some gaps in using advanced video models or analyzing task complexity. Reviewers generally appreciated the idea and experiments. Overall, the work tackles an important problem in multimodal models and offers practical solutions. The AC recommends acceptance, as the approach is promising and impactful, though there's room to explore dynamic scaling further.

**Additional Comments On Reviewer Discussion:**

The authors addressed most of concerns by adding detailed discussions on cross-attention models, introducing a token scale predictor with preliminary results, extending experiments to high-resolution tasks and SlowFast representations, and clarifying oracle metrics. Minor writing issues were also resolved. These updates and the experimental results strengthened the paper, and justified support for acceptance.

---

### Decision · Program_Chairs · 2025-01-22

Accept (Poster)